# PROJECTED NEURAL DIFFERENTIAL EQUATIONS FOR LEARNING CONSTRAINED DYNAMICS

## ABSTRACT

Neural differential equations offer a powerful approach for learning dynamics from data. However, they do not impose known constraints that should be obeyed by the learned model. It is well-known that enforcing constraints in surrogate models can enhance their generalizability and numerical stability. In this paper, we introduce *projected neural differential equations* (PNDEs), a new method for constraining neural differential equations based on projection of the learned vector field to the tangent space of the constraint manifold. In tests on several challenging examples, including chaotic dynamical systems and state-of-the-art power grid models, PNDEs outperform existing methods while requiring fewer hyperparameters. The proposed approach demonstrates significant potential for enhancing the modeling of constrained dynamical systems, particularly in complex domains where accuracy and reliability are essential.

## 1 INTRODUCTION

Numerical simulation of dynamical systems plays a pivotal role in science and engineering. Across fields, simulations extend the reach of scientific inquiry far beyond the limitations of direct observation and experimentation. However, as the scale and complexity of the systems being studied increases, the computational cost of traditional numerical methods can become prohibitive. Deep learning surrogates, enabled by the emergence of efficient architectures and specialized accelerator hardware, offer promising alternatives. By circumventing the computational limitations of conventional approaches, these approaches promise to further advance the use of simulations for scientific inquiry (Kochkov et al., 2021; Jumper et al., 2021; Pathak et al., 2022; Kovachki et al., 2023; Merchant et al., 2023; Azizzadenesheli et al., 2024).

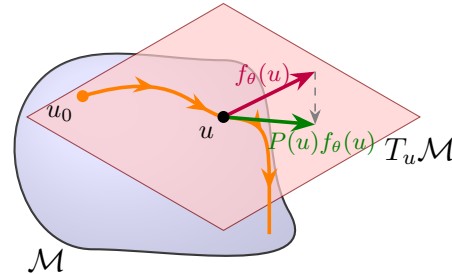

Figure 1: Schematic of projected neural differential equations (PNDEs). The vector field of the unconstrained NDE (red arrow) is projected to the tangent space $T_u\mathcal{M}$ of the constraint manifold $\mathcal{M}$. For initial conditions $u_0 \in \mathcal{M}$, solutions (orange) of the projected vector field (green arrow) remain on the manifold and thereby satisfy the constraints.

In power grid modeling, for example, simulations allow grid operators and researchers to examine the stability of the complex electrical networks that underpin modern life. However, the transition to renewable energy sources presents unprecedented challenges. As wind and solar generation increasingly contribute to the energy mix, grid operators must contend with distributed and highly variable production, alongside the loss of system inertia that was previously provided by conventional generators (Breithaupt et al., 2016; Milano et al., 2018). On the demand side, driven by electric vehicle adoption and the rapid growth of artificial intelligence, forecasts of peak demand have increased rapidly, further threatening grid stability (North American Electric Reliability Corporation (NERC), 2023). Unfortunately, existing methods for modeling grid dynamics are computationally infeasible at the scale and accuracy required for realistically modeling future scenarios dominated by renewable energy sources (Matevosyan et al., 2019). This problem is further exacerbated by legal mandates requiring grid operators to perform large-scale probabilistic simulations (European Network of Transmission System Operators for Electricity (ENTSO-e), 2018). By leveraging large

amounts of data to learn and predict grid dynamics, machine learning can enable grid operators to efficiently forecast system states, mitigate faults, and maintain stability in future power grids.

Among existing machine learning approaches, neural ordinary differential equations (NODEs) offer a promising approach for data-driven modeling of dynamical systems, combining the expressiveness of neural networks with the interpretability of differential equations (Chen et al., 2018). Building upon NODEs, universal differential equations (UDEs) offer a hybrid framework that integrates neural networks with differentiable physical models (Rackauckas et al., 2021). We refer to these approaches collectively as *neural differential equations* (NDEs). NDEs have demonstrated success across various scientific disciplines, including weather and climate modeling (Kochkov et al., 2024), robotics (Richards et al., 2021), epidemiology (Dandekar et al., 2020) and bioengineering (Narayanan et al., 2022). Despite excellent performance within the training data, NDEs often generalize poorly, for example, to new initial conditions or future times. Due to the iterative nature in which neural networks are applied within NDEs, inaccuracies and uncertainties are quickly amplified and can push the trajectory far outside the training distribution, where performance degrades.

Inductive biases play an important role in avoiding such issues. In scientific machine learning, for example, inductive biases can take the form of physical constraints, conservation laws, or other domain-specific knowledge that can enhance model performance and interpretability (Karniadakis et al., 2021). Physical constraints, in particular, are often incorporated as additional loss terms, for example, in physics-informed neural networks (Raissi et al., 2017a;b; 2019) and physics-informed operator learning (Li et al., 2021; Wang et al., 2021b; Goswami et al., 2022; White et al., 2023b). However, a variety of issues can arise with these *soft* constraints. For instance, the resulting composite loss can be expensive to evaluate, due to the derivatives in the physics loss, and challenging to minimize. Furthermore, a number of numerical issues are known to arise, such as worse conditioning, conflicting loss terms, inaccuracies in the approximated derivatives, and the existence of spurious solutions satisfying the constraints on the collocation points of the physics loss (Leiteritz and Pflüger, 2021; Krishnapriyan et al., 2021; Huang and Agarwal, 2023; Wang et al., 2021a; 2022; Rohrhofer et al., 2022). Yet perhaps the greatest drawback of soft constraints is that they provide no guarantees that the physical constraints will be satisfied at inference time; at best, they can only hope to guide the model weights towards physically plausible solutions during training. For certain safety-critical applications, where the ability to provably and robustly obey known physical constraints is crucial, this is inadequate. These critical drawbacks motivate stricter enforcement of the constraints, both during training and at inference time, in the form of *hard* constraints.

**Contribution.** In this paper, we introduce *projected neural differential equations* (PNDEs), a method for incorporating hard constraints in NDEs based on the projection of the vectors of the NDE's vector field onto tangent spaces of the constraint manifold (see Figure 1). Since the constraint manifold is defined by arbitrary algebraic constraints on the state of the system, our method can enforce diverse constraints, such as conservation laws, holonomic constraints, and even external forcings, all without requiring the system to be measured in specific (e.g., canonical) coordinates. Our method enforces constraints more accurately than existing methods, requires fewer hyperparameters to tune, and enhances the numerical stability and generalizability of the systems we study.

**Outline.** We motivate and derive the proposed method in Section 2, before surveying the existing literature and comparing to related works in Section 3. Finally, in Section 4, we demonstrate empirically on several challenging examples, including highly chaotic systems and elaborate power grid models, that our approach satisfies the constraints more accurately than state-of-the-art baselines, all while requiring fewer hyperparameters and permitting larger timesteps of the numerical solver.

## 2 PROJECTED NEURAL DIFFERENTIAL EQUATIONS

### 2.1 BACKGROUND

We first review some differential geometry definitions that are relevant to the proposed approach and refer the reader to (Lang, 1999; Lee, 2012; Boumal, 2024) for a more detailed treatment. The set of tangent vectors to a manifold $\mathcal{M}$ at a point $u \in \mathcal{M}$ is a vector space called the *tangent space* to $\mathcal{M}$ at $u$ and is denoted by $T_u\mathcal{M}$. The disjoint union of all the tangent spaces to $\mathcal{M}$ forms the *tangent bundle* $T\mathcal{M} = \{(u, v) \mid u \in \mathcal{M}, \ v \in T_u\mathcal{M}\}$ of $\mathcal{M}$. A *vector field* on a manifold $\mathcal{M}$ is a continuous

map $X : \mathcal{M} \to \mathrm{T}\mathcal{M}$ with the property that $X(u) \in \mathrm{T}_u\mathcal{M}$ for all $u \in \mathcal{M}$. The *integral curve* at $u$ of a vector field $X$ on $\mathcal{M}$ is the smooth curve $\gamma$ on $\mathcal{M}$ such that $\gamma(0) = u$ and $\gamma'(t) = X(\gamma(t))$ for all $t$ in the interval of interest. The integral curve $\gamma$ is simply the solution of a first-order ODE with right-hand side $X_{\gamma(t)}$ for $t$ in the interval of interest, so theorems for the existence, uniqueness, and smoothness of ODE solutions translate directly to integral curves of vector fields on manifolds.

## 2.2 PROBLEM SETTING

**Neural Differential Equations.** Assume we have an ambient $n$-dimensional Euclidean phase space $\mathcal{E}$, and consider the neural differential equation

$$\dot{u} = f_\theta(u(t), t) \qquad u(0) = u_0, \tag{1}$$

where $u : \mathbb{R} \to \mathcal{E}$, $u_0 \in \mathcal{E}$ is an initial condition, $\dot{u} = du/dt$, and $f_\theta$ is parameterized by a neural network with parameters $\theta \in \mathbb{R}^d$.

Given observations $\{t_i, u(t_i)\}_{i=1}^N$, the parameters $\theta$ can be optimized using stochastic gradient descent by integrating a trajectory $\hat{u}(t) = \mathrm{ODESolve}(u_0, f_\theta, t)$ and taking gradients of the loss $\mathcal{L}(u, \hat{u})$ with respect to $\theta$. Gradients of the ODESolve operation can be obtained using adjoint sensitivity analysis (sometimes called *optimize-then-discretize*) or via direct automatic differentiation of the solver operations (*discretize-then-optimize*) (Chen et al., 2018; Kidger, 2022).

Since any non-autonomous system can be equivalently represented as an autonomous system by adding time as an additional coordinate, for ease of notation we will drop the time-dependence in Equation (1) and consider only autonomous systems from now on, without loss of generality.

**Constraints.** Suppose that the state $u(t)$ of the system must be constrained to the set

$$\mathcal{M} = \{u \in \mathcal{E} \,;\, g(u) = 0\}, \qquad \text{where} \quad g\colon \mathcal{E} \to \mathbb{R}^m, \;\; g : u \mapsto \big(g_1(u), \ldots, g_m(u)\big), \tag{2}$$

given $m < n$ independent explicit algebraic constraints $g_i(u) = 0$. Assuming that the Jacobian $Dg(u)$ exists and has full rank for all $u \in \mathcal{M}$, the Regular Level Set Theorem (Lee, 2012, #5.14) implies that $\mathcal{M}$ is a smooth $(n-m)$-dimensional embedded submanifold of $\mathbb{R}^n$. To satisfy the constraints $g(u) = 0$, we seek solutions of Equation (1) that are constrained to the embedded submanifold $\mathcal{M}$, which we refer to as the *constraint manifold*. In other words, we seek integral curves on $\mathcal{M}$.

## 2.3 PROPOSED APPROACH

**Proposed Approach.** We want to learn $f_\theta : \mathcal{M} \to \mathrm{T}\mathcal{M}$ as a smooth vector field on the embedded submanifold $\mathcal{M}$ with local defining function $g$. We realize this by first parameterizing a smooth extension $\bar{f}_\theta\colon \mathcal{E} \to \mathcal{E}$ of $f_\theta$ in Euclidean space using a neural network, and then projecting $\bar{f}_\theta(u)$ using a suitable projection operator $\mathrm{Proj}_u\colon \mathcal{E} \to \mathrm{T}_u\mathcal{M}$ to obtain the desired tangent vector,

$$f_\theta(u) = \mathrm{Proj}_u\big(\bar{f}_\theta(u)\big) \in \mathrm{T}_u\mathcal{M}. \tag{3}$$

Since $\mathcal{M}$ is a smooth embedded submanifold of $\mathcal{E}$, the restriction $f_\theta = \bar{f}_\theta|_\mathcal{M}$ is also smooth. Note that the projection acts fibrewise (i.e. acting "point-by-point"); for each $u \in \mathcal{M}$, it projects the vector $\bar{f}_\theta(u)$ at $u$ onto $\mathrm{T}_u\mathcal{M}$, as opposed to projecting directly the full vector field $\bar{f}_\theta$ onto $\mathrm{T}\mathcal{M}$.

**Construction of the projection operator.** We now derive a suitable projection operator, following the approach of Boumal (2024). Given an inner product $\langle \cdot, \cdot \rangle$ and induced norm $\|\cdot\|$ on $\mathcal{E}$, the differential $Dg(u)$ and its adjoint $Dg(u)^*$ define linear maps

$$Dg(u)[v] = \big(\langle \nabla g_1(u), v\rangle, \ldots, \langle \nabla g_m(u), v\rangle\big), \qquad Dg(u)^*[\alpha] = \sum_{i=1}^m \alpha_i \nabla g_i(u). \tag{4}$$

Denoting the orthogonal projection from $\mathcal{E}$ to $\mathrm{T}_u\mathcal{M}$ by $\mathrm{Proj}_u : \mathcal{E} \to \mathrm{T}_u\mathcal{M}$, we can uniquely decompose any vector $v \in \mathcal{E}$ into its components parallel and perpendicular to $\mathrm{T}_u\mathcal{M}$,

$$v = \mathrm{Proj}_u(v) + Dg(u)^*[\alpha]. \tag{5}$$

The coefficients $\alpha$ are obtained as the solution of the least-squares problem

$$\alpha = \underset{\alpha \in \mathbb{R}^m}{\arg\min} \|v - Dg(u)^*[\alpha]\|^2 = \big(Dg(u)^*\big)^\dagger[v], \tag{6}$$

where $(\cdot)^\dagger$ denotes the Moore-Penrose pseudoinverse. Substituting Equation (6) into Equation (5), we obtain a formula for the orthogonal projection from $\mathcal{E}$ to $\mathrm{T}_u\mathcal{M}$,

$$\mathrm{Proj}_u(v) = v - \mathrm{D}g(u)^* \left[ \left(\mathrm{D}g(u)^*\right)^\dagger [v] \right]. \tag{7}$$

**Projected Neural Differential Equation.**    Projecting the NDE in Equation (1) using the projection operator defined in Equation (7) gives the following *Projected Neural Differential Equation* (PNDE),

$$\dot{u} = \mathrm{Proj}_u(f_\theta(u)) \in \mathrm{T}_u\mathcal{M}, \tag{8}$$

which is a differential equation on the constraint manifold $\mathcal{M}$.

**Proposition 1.** *Solutions to the PNDE 8 with $u_0 \in \mathcal{M}$ satisfy $g(u(t)) = 0$ for all $t \geq 0$.*

*Proof.* We note that

$$\frac{dg(u(t))}{dt} = Dg(u)[\dot{u}] = Dg(u)\left[\mathrm{Proj}_u(f_\theta(u))\right] = 0 \tag{9}$$

since $\mathrm{Proj}_u(f_\theta(u)) \in \mathrm{T}_u\mathcal{M} = \ker \mathrm{D}g(u)$. As a result, $g(u(t))$ remains constant in time and satisfies $g(u(t)) = g(u(0)) = 0$ for all $t \geq 0$ since $u_0 \in \mathcal{M}$.                □

## 3    RELATED WORK

**Hard Constraints.**    A number of projection-based approaches have been proposed to constrain the outputs of deep learning models. Harder et al. (2024) design constraint layers to ensure mass and energy conservation during downscaling (i.e., super-resolution) of climate model fields from low to high resolution. Negiar et al. (2022) and Chalapathi et al. (2024) add a differentiable implicit constraint layer for finding optimal linear combinations of learned basis functions that approximately satisfy differential constraints at collocation points. Jiang et al. (2020) and Duruisseaux et al. (2024) use a spectral projection layer to enforce linear differential constraints efficiently in Fourier space. Hard constraints have also been use to take advantage of structured and well-understood dynamical systems, for example, the divergence-free property of incompresible fluid flows (Richter-Powell et al., 2022; Mohan et al., 2023; Xing et al., 2024) and the symplectic structure of Hamiltonian systems (Lutter et al., 2019; Greydanus et al., 2019; Zhong et al., 2020; Jin et al., 2020; Burby et al., 2020; Cranmer et al., 2020; Sæmundsson et al., 2020; Valperga et al., 2022; Duruisseaux et al., 2023a;b).

The work of Finzi et al. (2020) is especially relevant to this paper. They use explicit algebraic constraints to enable the learning of Hamiltonian dynamics in Cartesian coordinates, where the Hamiltonian is "simpler" and therefore easier to learn than in the original generalized coordinates. They add constraints to the Hamiltonian via Lagrange multipliers, and use a variational approach to derive the corresponding constrained equations of motion. The result is the standard Hamiltonian equations of motion, multiplied by a projection operator that enforces the constraints in a manner that is consistent with the underlying Hamiltonian structure. While derived from an entirely different perspective, our approach can be understood as a principled generalization of their method beyond the idealized Hamiltonian setting to a much more general class of NDEs. In this paper, for example, we include tests on a damped pendulum system, which is not compatible with their method.

**Stabilized Neural Differential Equations.**    White et al. (2023a), whose work is most closely related to ours, constrain NDE solutions using stabilized neural differential equations (SNDEs), given by

$$\dot{u} = f_\theta(u) - \gamma F(u)g(u), \tag{10}$$

where $\gamma \geq 0$ is a scalar, $F : \mathbb{R}^n \to \mathbb{R}^{n \times m}$ is a stabilization matrix, and $g : \mathbb{R}^n \to \mathbb{R}^m$ is the same constraint function we defined in Equation (2). Under mild conditions on $\gamma$ and $F$, the stabilized NDE (10) admits all solutions of the original NDE (1) while rendering the constraint manifold provably asymptotically stable. In other words, the stabilized NDE (10) is able to learn the correct ground truth dynamics while guaranteeing that the constraints are satisfied.

To understand how SNDEs relate to our proposed method, we first provide the intuition behind Equation (10). The stabilization term (i.e. the second term) equals zero on the manifold, since

$g(u) = 0$ there by definition. Hence, the stabilization is only "activated" when a trajectory leaves the constraint manifold, i.e., when the constraints have already been violated. The effect of the stabilization is to "nudge" such a trajectory "back towards $\mathcal{M}$", i.e., back towards satisfying the constraints, with the strength of the nudge determined by the stabilization parameter $\gamma$. Larger values of $\gamma$ lead to a stronger nudge and stricter enforcement of the constraints. However, excessively large values of $\gamma$ may require the numerical solver to stop more frequently, with the system eventually becoming stiff. The key difference with our method is that SNDEs continuously correct violations of the constraints, while our method prevents such violations from occurring in the first place. Accordingly, we can expect our method to yield a stronger enforcement of the constraints. We will use SNDEs as a baseline for comparison throughout this paper.

**Continuous Normalizing Flows on Riemannian Manifolds.** A large body of related work has studied neural ODEs on Riemannian manifolds, primarily in the context of continuous normalizing flows on non-Euclidean geometries. In contrast to our approach, they typically deal with prototypical Riemannian manifolds, such as spheres and tori, and are not easily adapted to more general manifolds. For example, the approach of Lou et al. (2020) requires a local chart for the manifold, the derivation of which is likely to be analytically intractable for embedded submanifolds with arbitrary and nonlinear local defining functions. Similarly, the approaches of Bose et al. (2020) and Rezende et al. (2020) require an expression for the exponential map, while Mathieu and Nickel (2020) uses a custom geodesic distance layer, none of which are easily applied in our setting.

**Optimization and Numerical Integrators on Manifolds.** The use of projections is widespread in optimization on manifolds (Absil et al., 2008; Boumal, 2024), where it is used to project iterates to the constraint manifold and/or gradients onto appropriate tangent spaces (as in our approach). The idea of projecting gradients onto tangent spaces of the constraint set dates back at least to Luenberger (1972; 1973), and is still commonly used. For instance, in Riemannian optimization, Euclidean gradients are typically replaced by Riemannian gradients, which are their orthogonal projections onto tangent spaces. Explicit formulas and efficient projection algorithms have been derived and used for manifolds of greater practical interest (Edelman et al., 1998; Manton, 2002; Absil et al., 2008; Wen and Yin, 2010; Absil and Malick, 2012; Hauswirth et al., 2016; Zhang and Sra, 2016; Oviedo León and Dalmau-Cedeño, 2019; Oviedo León et al., 2021). Where these methods differ crucially from our approach is that they typically seek to constrain the weights or optimization variables themselves to a given matrix manifold, while our aim is to constrain trajectories of an NDE, the weights of which are not themselves directly constrained. Regarding numerical integration on manifolds, the available literature is much less extensive (Hairer, 2011), and most approaches restrict themselves to Lie groups (very structured Riemannian manifolds) and Hamiltonian or mechanical systems, in order to obtain practical algorithms (Iserles et al., 2000; Christiansen et al., 2011; Celledoni et al., 2014; 2022). Some of these approaches are based on projection, similarly to optimization on manifolds, while others use local parametrizations or Lagrange multipliers. Riemannian optimization algorithms have also been obtained by simulating dynamical systems using numerical integrators constrained to the relevant Riemannian manifolds (Alimisis et al., 2020; Tao and Ohsawa, 2020; Lee et al., 2021; França et al., 2021; Duruisseaux and Leok, 2022a;b;c; 2023).

## 4 Experiments

We now test the proposed approach on several challenging examples, including highly chaotic systems and elaborate power grid models. The details of the data generation, model setup, and training procedure, are provided in Appendices A and B. We evaluate each model's predicted trajectory $\hat{u}(t)$ versus the ground truth trajectory $u(t)$ using the usual relative error $E(t) = \|u(t) - \hat{u}(t)\|_2 / \|u(t)\|_2$. For chaotic dynamics, we instead use the bounded relative error $E(t) = \|u(t) - \hat{u}(t)\|_2 / (\|u(t)\|_2 + \|\hat{u}(t)\|_2)$, which approaches 1 as trajectories become orthogonal. All test statistics are averaged over 100 unseen initial conditions.

### 4.1 Conservation Laws

The Fermi–Pasta–Ulam–Tsingou (FPUT) lattice system (Fermi et al., 1955) describes a vibrating string with nonlinear dynamics. The displacement $x_j$ of a segment $j$ with mass $m$ is governed by the second order equation,

$$m\ddot{x}_j = k(x_{j+1} + x_{j-1} - 2x_j)\left[1 + \alpha(x_{j+1} - x_{j-1})\right], \tag{11}$$

where the first factor is simply Hooke's law with spring constant $k$, and the second factor introduces a nonlinearity governed by the parameter $\alpha$.

In the case of a circular lattice with $N$ segments and periodic boundary conditions $x_i = x_{i+N}$, the system conserves energy

$$E(x, \dot{x}) = \sum_{j=1}^{N} \left[ \frac{1}{2} m \dot{x}_j^2 + \frac{1}{2} k (x_{j+1} - x_j)^2 + \frac{1}{3} \alpha (x_{j+1} - x_j)^3 \right], \tag{12}$$

where $x = (x_1, \ldots, x_N)$ and $\dot{x} = (\dot{x}_1, \ldots, \dot{x}_N)$. We convert Equation (11) into a first order ODE with state $u = (x, v) \in \mathbb{R}^{2N}$ where we introduce the variable $v := \dot{x}$. We suppose that we possess prior physical knowledge of the conservation law in Equation (12), which we enforce as the constraint

$$g(u) = E(u) - E(u_0) = 0, \tag{13}$$

where $g : \mathbb{R}^{2N} \to \mathbb{R}$ and $u_0$ is the initial condition.

Figure 2 compares PNDEs with SNDEs and an unconstrained NDE on the FPUT system with $N = 128$ segments. The unconstrained NDE is unstable and diverges for long rollouts (longer than the training trajectories and significantly longer than an individual training sample). In contrast, all of the constrained models are stable, highlighting the value of enforcing known constraints in NDE models. PNDEs and SNDEs demonstrate similar ability to predict the evolution of the system state, although they both eventually depart from the ground truth trajectory of this nonlinear system. The PNDE model enforces the constraints several orders of magnitude more accurately than both SNDE models, despite requiring significantly fewer evaluations of the the right-hand side function $f_\theta$.

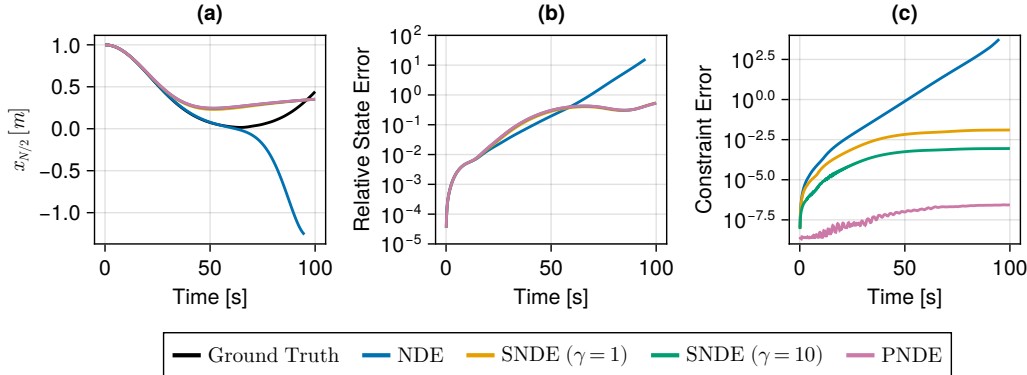

Figure 2: Performance of PNDEs compared to SNDEs and an unconstrained NDE for the Fermi–Pasta–Ulam–Tsingou (FPUT) lattice system (Fermi et al., 1955) with $N = 128$ segments. Panel (a) shows the ground truth as well as the NDE, SNDE, and PNDE predictions of the system state. Panel (b) shows the relative error. Panel (c) shows the error in the constrained quantity. Note that PNDE and SNDE results in (a) and (b) are visually indistinguishable.

## 4.2 SIMPLIFYING COMPLEX LEARNING TASKS USING EXPLICIT CONSTRAINTS

We now consider an $N$-pendulum system, consisting of $N$ simple pendulums connected end-to-end, as illustrated on the right for $N = 3$. $N = 1$ corresponds to the canonical simple pendulum, $N = 2$ yields the well-known chaotic double pendulum system, and higher values of $N$ yield systems with progressively more complex and chaotic dynamics. For realism, and in contrast to much of the related literature, we include damping proportional to the relative angular velocity of each pendulum bob; this could represent, for example, friction in the hinges of the pendulum.

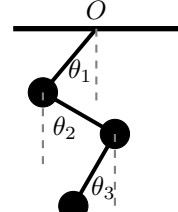

Figure 3: Illustration of a 3-pendulum.

Assuming that each bob has a mass of $1\,\text{kg}$ and each arm has a length of $1\,\text{m}$, the equations of motion of the $i$-th bob are then given by

$$\sum_{j=1}^{N} a(i, j) \left[ \ddot{\theta}_j \cos(\theta_i - \theta_j) + \dot{\theta}_j^2 \sin(\theta_i - \theta_j) \right] + b(\dot{\theta}_i - \dot{\theta}_{i-1}) + (N - i + 1)\mathfrak{g} \sin \theta_i = 0, \tag{14}$$

where $\theta_i$ is the angle that the $i$-th arm makes with the vertical, $b$ is the coefficient of friction, $a(i, j) = N - \max(i, j) + 1$, and $\mathfrak{g} = 9.81 \, \mathrm{m \, s^{-1}}$ is acceleration due to gravity.

Given observations of this system, a reasonable approach would be to learn an NDE directly approximating Equation (14), which is in generalized coordinates. Alternatively, and analogously to the approach taken by Finzi et al. (2020) for Lagrangian and Hamiltonian systems, we can first convert to Cartesian coordinates, where the equations of motion may be easier to learn. In Cartesian coordinates, constraints arise naturally from the rigidity of the pendulum arms, i.e., the length of each arm should remain constant in time. Hence, given the position $q_i = (x_i, y_i)$ of the $i$-th bob in Cartesian coordinates, we can write down $N$ holonomic constraints on the system,

$$g_i^{(pos)}(q) = ||q_i - q_{i-1}||^2 - 1 = 0, \quad i = 1, \ldots, N, \tag{15}$$

where $q = (q_1, \ldots, q_N)$ and $q_0 = (0, 0)$ is the origin. We can differentiate these constraints to obtain $N$ additional conjugate constraints on the velocity,

$$g_i^{(vel)}(q) = (q_i - q_{i-1}) \cdot (\dot{q}_i - \dot{q}_{i-1}) = 0, \quad i = 1, \ldots, N. \tag{16}$$

Collecting the position and velocity constraints into a vector, we obtain $2N$ constraints for the $N$-pendulum in Cartesian coordinates:

$$g(q) = \left[ g_1^{(pos)}(q), \ldots, g_N^{(pos)}(q), \ g_1^{(vel)}(q), \ldots, g_N^{(vel)}(q) \right] = 0, \tag{17}$$

where $g : \mathbb{R}^{4N} \to \mathbb{R}^{2N}$. We note that the generalized coordinates of Equation (14) satisfy these constraints automatically.

Figure 4 demonstrates that learning constrained dynamics in Cartesian coordinates produces significantly better predictions than learning the same dynamics expressed in generalized coordinates, where the constraints are satisfied automatically. The benefits of learning in Cartesian coordinates with constraints becomes especially pronounced as the size and complexity of the system increases. Amongst constrained models, PNDEs enforce the constraints several orders of magnitude more accurately than SNDEs and do not require tuning of the stabilization parameter $\gamma$. Unconstrained NDEs in Cartesian coordinates are numerically unstable, again highlighting the importance of constraints.

### 4.3 POWER GRIDS

The IEEE test systems are standardized power system models widely used in electrical engineering research and education (Christie, 1999). Representing key features of power grids such as topologies and operating conditions, these models enable researchers to benchmark algorithms and offer a common framework for analyzing power flow, stability, and reliability in electrical networks. We will study the dynamics of two test systems in particular: the IEEE 14-Bus system (see Figure 5, right) and the IEEE Reliability Test System 1996 (hereafter RTS-96) (Grigg et al., 1999). We adapt the IEEE test cases to represent future grid dynamics by replacing conventional generators with renewable energy sources. Our renewables-based grids consist of three types of buses, or nodes: (1)

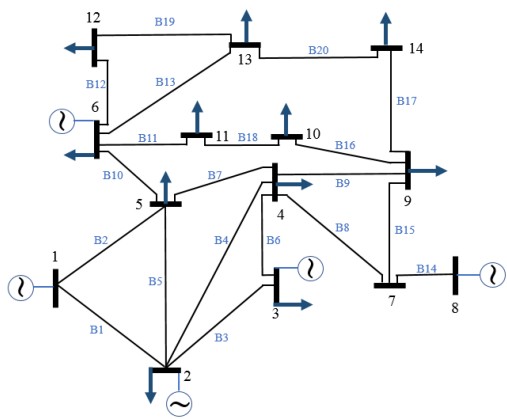

Figure 5: The IEEE 14-Bus system. Circles represent generators and arrows represent consumers. Diagram extracted from Leon et al. (2020).

renewable energy sources, (2) loads (that is, consumers), and (3) slack buses (a constant voltage bus representing, for example, a large power plant or a connection to another grid).

The state of each bus in the grid is determined by a complex voltage, the real and imaginary part of which we learn separately. That is, for a grid consisting of $N$ buses, we learn the coupled dynamics of $2N$ voltages. Normally, a power grid operates at a steady-state equilibrium, called its *operation point*. We apply random perturbations to each grid (see Appendix A) and learn the dynamics from the resulting transients as it returns to its operation point.

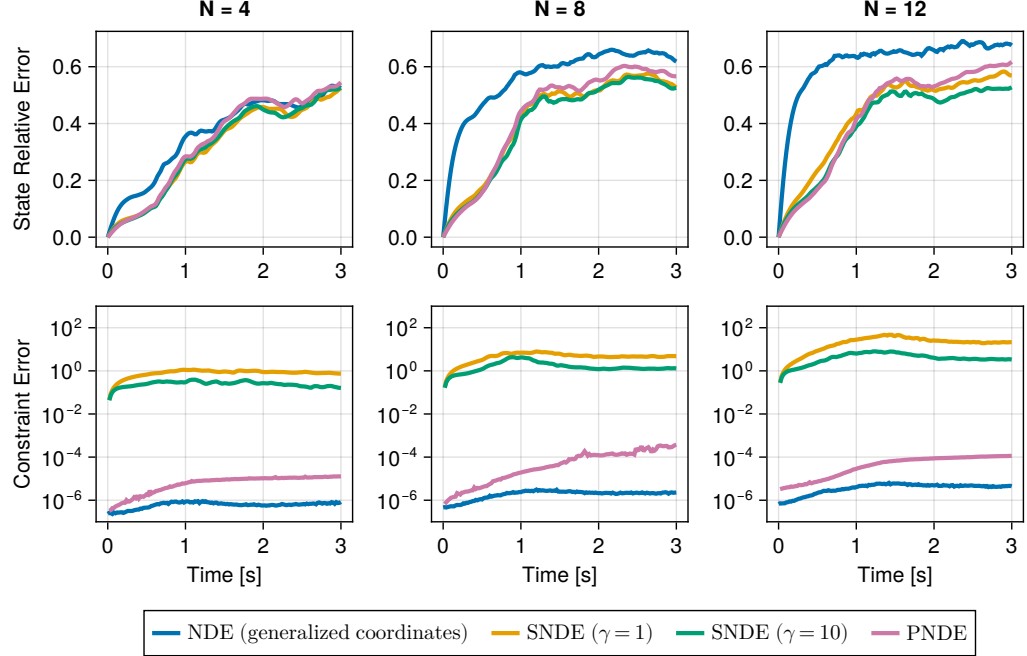

Figure 4: Damped $N$-pendulum system. We compare the ability of an NDE to learn the dynamics of the damped $N$-pendulum system in generalized coordinates, where the constraints are satisfied automatically, versus Cartesian coordinates, where we explicitly enforce the constraints. **Top:** Across all systems, constrained models in Cartesian coordinates better predict the future evolution of the system's state. The difference in performance becomes more pronounced as $N$ is increased and the corresponding equations of motion become more complex in generalized coordinates. **Bottom:** PNDEs enforce the constraints exactly, while SNDEs incur large errors, especially for larger values of $N$. For SNDEs, smaller values of $\gamma$ than those shown here do not enforce the constraints, while larger values are computationally prohibitive due to stiffness. In generalized (angular) coordinates, the constraints are satisfied automatically by the choice of coordinate system. Nevertheless, the results show that converting to Cartesian coordinates and adding constraints leads to significantly better generalization. Unconstrained NDEs in Cartesian coordinates (not shown) are numerically unstable, highlighting the necessity of imposing explicit constraints via SNDEs or PNDEs.

In our grid models, loads are modeled using *PQ-buses*, meaning that the complex power at the PQ-bus $j$ is set to a fixed, constant value given by

$$P_j + \mathrm{i}Q_j = v_j \cdot i_j^* \tag{18}$$

where $P_j$ and $Q_j$ are the fixed active and reactive power, respectively, $v_j$ is the complex voltage, and $i_j$ is the complex current (Machowski et al., 2008). These fixed power values will be the constraints in our experiments. The complex currents in Equation (18) are determined from the voltages by $i = \mathrm{LY} \times v$, where $i \in \mathbb{C}^N$, $v \in \mathbb{C}^N$, and the admittance Laplacian $\mathrm{LY} \in \mathbb{C}^{N \times N}$ describes the line admittances, which are well understood and assumed here to be known. Given $N$ total buses and $M$ PQ-buses, the $2M$ constraints (one real and one imaginary for each PQ-bus) are given by

$$g(u) = [P_1 - \mathrm{Re}(v_1 \cdot i_1^*), Q_1 - \mathrm{Im}(v_1 \cdot i_1^*), \ldots, P_M - \mathrm{Re}(v_M \cdot i_M^*), Q_M - \mathrm{Im}(v_M \cdot i_M^*)] = 0, \tag{19}$$

where $u = (\mathrm{Re}(v_1), \mathrm{Im}(v_1), \ldots, \mathrm{Re}(v_N), \mathrm{Im}(v_N)) \in \mathbb{R}^{2N}$ and $g : \mathbb{R}^{2N} \to \mathbb{R}^{2M}$. For the IEEE 14-bus system, $N = 14$ and $M = 9$, while for the IEEE RTS-96 system, $N = 74$ and $M = 40$.

Table 1 shows the mean-squared error for both IEEE test systems, calculated over the entire test set, and Figure 6 shows the relative error as a function of time, along with some illustrative trajectories.

Unconstrained NDEs incur large errors in the PQ-bus power constraint. For SNDEs, increasing the stabilization hyperparameter $\gamma$ enforces the constraints more accurately, but larger values lead to stiffness and, ultimately, prohibitively expensive simulations. Meanwhile, PNDEs enforce the constraints close to machine precision and several orders of magnitude more accurately than the best SNDEs without stiffness.

Table 1: MSE of the predicted voltage $V$ (all buses) and power constraint (PQ-buses only), along with the number of function evaluations required by the adaptive ODE solver. For both systems, the constrained models predict the voltages at least as well as the NDE. Unsurprisingly, the NDE incurs large errors in the PQ-bus power. SNDEs enforce the constraints more accurately for larger values of $\gamma$, but the largest values lead to stiffness, requiring many more function evaluations. PNDEs efficiently enforce the constraints near machine precision.

| Model | IEEE 14-Bus | | | IEEE RTS-96 | | |
|---|---|---|---|---|---|---|
| | $V$ ($\times 10^{-4}$) | PQ Power | $f_\theta$ Evals | $V$ ($\times 10^{-3}$) | PQ Power | $f_\theta$ Evals |
| NDE | $3.4\pm1.1$ | $(2.2\pm0.4)\times10^{-1}$ | 129 | $3.3\pm0.8$ | $4.0\pm0.2$ | 411 |
| SNDE ($\gamma=0.1$) | $3.0\pm1.0$ | $(1.1\pm0.2)\times10^{-1}$ | 135 | $3.1\pm0.7$ | $1.86\pm0.14$ | 411 |
| SNDE ($\gamma=1$) | $2.7\pm1.0$ | $(1.1\pm0.8)\times10^{-2}$ | 141 | $2.8\pm0.7$ | $(1.7\pm0.4)\times10^{-1}$ | 411 |
| SNDE ($\gamma=10$) | $2.7\pm1.0$ | $(0.3\pm1.3)\times10^{-3}$ | 273 | $2.5\pm0.6$ | $(5.0\pm7.0)\times10^{-3}$ | 411 |
| SNDE ($\gamma=100$) | $2.7\pm1.0$ | $(0.0\pm1.6)\times10^{-4}$ | 1899 | $2.5\pm0.6$ | $(0.0\pm9.0)\times10^{-4}$ | 2199 |
| PNDE | $2.7\pm1.0$ | $\mathbf{(0.0\pm0.7)\times10^{-6}}$ | 135 | $2.5\pm0.6$ | $\mathbf{(0.0\pm1.7)\times10^{-6}}$ | 405 |

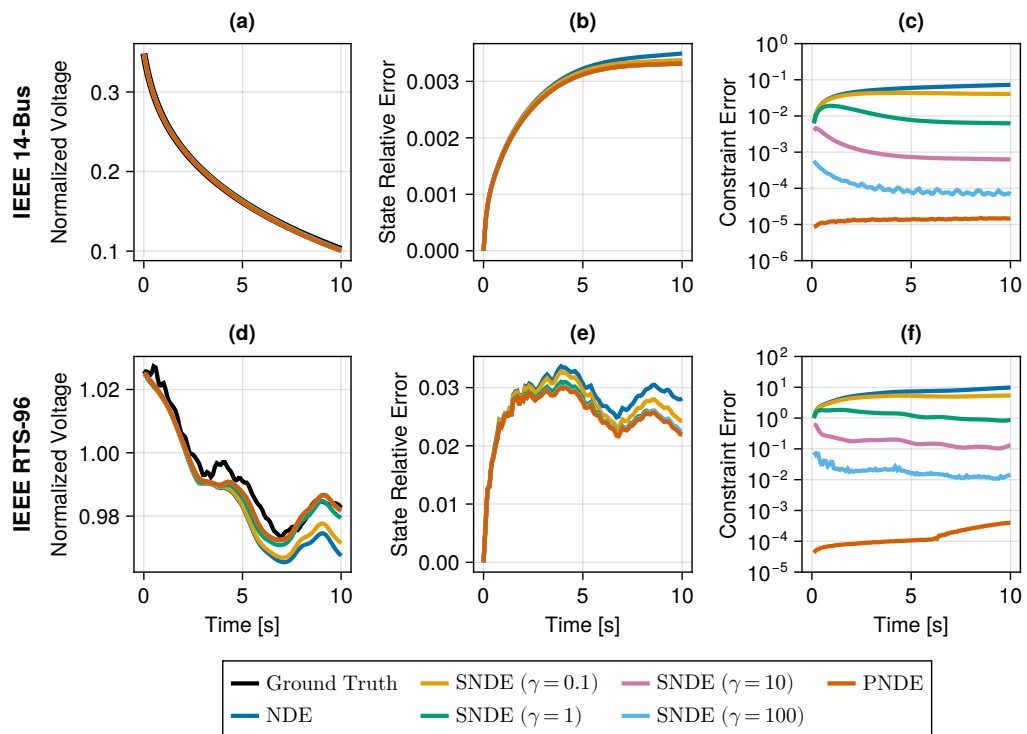

Figure 6: Test statistics for the IEEE 14-Bus System (top) and IEEE RTS-96 System (bottom). Panels (a) and (d) show a representative voltage for a single trajectory. All models accurately capture the shape of this voltage for the 14-bus system, but appear to struggle with the high frequency components of the RTS-96 system. Panels (b) and (e) show the relative error of the voltages at all buses, calculated over the entire test set. Constrained models outperform unconstrained models, with the gap growing over time. Panels (c) and (f) show the relative error in the power at the PQ-buses, with the PNDE outperforming the best SNDEs by several orders of magnitude.

## 5 CONCLUSION

We have introduced projected neural differential equations (PNDEs), a new method for enforcing arbitrary algebraic constraints in neural differential equation models. Compared to state-of-the-art baselines, our approach requires fewer hyperparameters and enforces the constraints several orders of

magnitude more accurately. In experiments on complex nonlinear and chaotic systems, as well as two challenging examples based on state-of-the-art power grid models, PNDEs are efficient, numerically stable, and generalize well to unseen data. We have also demonstrated the use of PNDEs to simplify and significantly improve the learning of complex dynamical systems using explicit constraints. Many promising directions remain for future research. For example, graph neural networks are also compatible with our method and a natural fit for modeling grid dynamics, thereby enabling critical out-of-distribution tests such as changing grid topologies.

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

## A  DATASETS

For each system, we generate ground truth trajectories from random initial conditions. The exact number of initial conditions used for training, validation, and testing, as well as the duration of each trajectory and the timestep used, is determined based on the individual characteristics of each system, and summarized in Table 2. For training and validation, each trajectory is split into contiguous segments consisting of 4 timesteps, which are then randomized and combined along the batch dimension. The first observation in each segment is the initial condition for the prediction.

With the exception of the power grid experiments, all trajectories are generated using the 9(8) explicit Runge-Kutta algorithm due to Verner (2010), implemented in DifferentialEquations.jl (Rackauckas and Nie, 2017) as `Vern9`, with absolute and relative solver tolerances of $10^{-12}$.

The power grid models, which are differential-algebraic equations (DAEs), are implemented in the Julia package PowerDynamics.jl (Plietzsch et al., 2022). Given a random initial condition, we then generate a ground truth trajectory by solving the DAE for 10 seconds using a 4th order L-stable Rosenbrock method (Hairer and Wanner, 1996), implemented in the Julia package DifferentialEquations.jl (Rackauckas and Nie, 2017) as `Rodas4`.

Table 2: Summary of datasets and hyperparameters

| System | $n_{\text{train}}$ | $n_{\text{valid}}$ | $n_{\text{test}}$ | $T$ | $dt$ | Hidden Layers | Hidden Width | Epochs |
|---|---|---|---|---|---|---|---|---|
| FPUT | 120 | 60 | 100 | 25 | 0.1 | 3 | 512 | 1000 |
| $N$-pendulum | 160 | 80 | 100 | 10 | 0.02 | 3 | 1024 | 5000 |
| IEEE 14-Bus | 100 | 50 | 100 | 10 | 0.1 | 3 | 512 | 5000 |
| IEEE RTS-96 | 100 | 50 | 100 | 10 | 0.1 | 3 | 512 | 5000 |

### A.1  INITIAL CONDITIONS

**Fermi–Pasta–Ulam–Tsingou System.**  Given a string defined on the range $x \in [0, 1)$, we generate a symmetric initial displacement according to $x_0 \sim \exp\left(-(x - \mu)^2/\sigma^2\right)$, where $\mu = 0.5$ and $\sigma \sim \text{Unif}(0.1, 0.3)$. We release the string from rest, i.e., we set $\dot{x}_0 = 0$.

**$N$-Pendulum.**  The initial displacements of the pendulum arms are drawn independently from $\text{Unif}(0, 2\pi)$ and the initial velocities are drawn from $\text{Unif}(-1, 1)$.

**Power Grids.**  Given a power grid at its operation point (i.e., a stable equilibrium with all variables at their desired steady-state values) we obtain perturbed initial conditions using the ambient forcing method of Büttner et al. (2022), which ensures that the perturbed state satisfies the PQ-bus constraints. The transient behavior of the grid following the perturbation produces rich and complex dynamics that we aim to emulate using NDEs.

## B  TRAINING

All models are trained on NVIDIA H100 GPUs. We minimize the mean squared error loss using the AdamW optimizer (Loshchilov and Hutter, 2018) with a constant learning rate of $10^{-3}$, weight decay of $10^{-4}$, and a batch size of 1024. All systems are modeled using standard multilayer perceptrons with GELU activation functions to encourage smoothness. The number of hidden layers, hidden width, and training epochs are summarized in Table 2. All models are initially trained without constraints, and subsequently fine-tuned with the constraints for 100 epochs.

During training, all models are solved using the 5(4) explicit Runge-Kutta algorithm of Tsitouras (2011), implemented in DifferentialEquations.jl (Rackauckas and Nie, 2017) as `Tsit5`, with absolute and relative tolerances of $10^{-4}$. Gradients of ODE solutions are obtained using the adjoint sensitivity method (Pontryagin, 1962; Chen et al., 2018), implemented in SciMLSensitivity.jl as `BacksolveAdjoint` (Rackauckas et al., 2021). For testing, we use absolute and relative tolerances of $10^{-6}$.

