# OpenReview forum: "Projected Neural Differential Equations for Learning Constrained Dynamics"
_ICLR.cc/2025/Conference — ICLR 2025 Conference Withdrawn Submission_

### Official Review · Reviewer_MdTC · 2024-10-23

**Soundness:** 2
**Presentation:** 4
**Contribution:** 3
**Rating:** 5
**Confidence:** 4

**Summary:**

The paper introduces Projected Neural Differential Equations (PNDEs), a novel method designed to incorporate known constraints into neural differential equations. Specifically, PNDEs project the vectors of the vector field onto the tangent spaces of the manifold defined by known constraints. This allows for the enforcement of various constraints without the need for specific coordinate systems. The paper provides empirical evidence through experiments showing that PNDEs satisfy constraints more accurately than state-of-the-art baselines.

**Strengths:**

1. The paper presents an approach to integrating known constraints into neural differential equations, which is a relatively unexplored area in the field of machine learning and dynamical systems. The introduction of the projection method to enforce constraints on the vector field is innovative.

2. The empirical results demonstrate that PNDEs outperform existing methods in terms of accuracy and stability. The experiments conducted on challenging examples, including chaotic systems and power grid models, further validate the robustness of the proposed method.

3. The paper is well structured and clearly written, making complex concepts accessible to readers.

**Weaknesses:**

1. There is a potential ambiguity in the notation used in the paper. Specifically, the definition of $\mathcal{E}$ is lacking; is it $\mathbb{R}^n$? The notations $f_{\theta}$ and $\bar{f}_{\theta}$ in equations 1, 3, and 8 are not entirely consistent.

2. The paper primarily focuses on known constraints. In many cases—such as the first example presented—it shows that if the constraints are known, the corresponding total differential equation can be determined.

3. The effect of the ODE solver is not discussed in this paper. Ideally, the constraints should be preserved exactly, as stated by Proposition 1 in the paper. However, the numerical results indicate that they are only preserved approximately, albeit with small errors. These discrepancies may arise from numerical errors introduced by the ODE solver.

4. The setup of the first example differs from that in Section 2. Is it possible to explicitly write out the manifold for the example?

5. The empirical results demonstrate that the proposed PNDEs outperform existing methods in terms of the consistent error. However, the improvement in terms of prediction error is less pronounced. I am not certain whether the ability to preserve the given constraints is the most critical indicator. If we aim to improve this, the simplest and most straightforward approach would be to project the predicted state onto the known manifold during the prediction phase.

**Questions:**

1. Could the paper provide a detailed formula of computing the "Relative State Error" and "Constraint Error"?

2. Could the paper explain in detail how the trajectory of the figure was selected?

3. What does the vertical axis of the leftmost subgraph in Figure 2 mean？

4. Section 4.2: Why does NDE using generalized coordinates that satisfy constraints perform poorly, and does this mean that preserving constraints cannot directly indicate better predictions? Can the results highlight the importance of constraints?

5. Section 4.3: Do we know the governing function for this example, or are the dynamics learned from the given data? Additionally, the statement 'apply random perturbations to each grid (see Appendix A) and learn the dynamics from the resulting transients' feels unclear. Could the paper clarify this? Also, what are the sizes of the training and test datasets used for this example?

6. For all examples, the paper only presents the state error for a single test trajectory. Could the paper provide more comprehensive quantitative evaluations?

7. The assumption of known constraints may be too restrictive. It would be helpful to discuss scenarios where the constraints are known, but the governing function is unknown and data is provided.

---

### Official Review · Reviewer_TExi · 2024-11-01

**Soundness:** 2
**Presentation:** 2
**Contribution:** 1
**Rating:** 5
**Confidence:** 3

**Summary:**

This paper presents a projection-based approach to ensure hard constraint satisfaction in constrained dynamics, which is important for several real-world applications. However, several concerns outlined in the weaknesses section remain, and further investigation of these limitations is needed to improve its practical applicability.

**Strengths:**

Constrained dynamics are present in real-world problems, and existing NODEs that do not consider these constraints run the risk of failing to satisfy them. In contrast, the proposed method achieves hard constraint satisfaction through projection.

**Weaknesses:**

1.	The methodology is based on the assumption that the analytic form of the constraint function $g$ is known, which seems impractical. In real situations where the dynamics are unknown and the states $u$ are given only by the data, there are often many cases where the analytic form of $g$ is also unknown.
2.	What is the difference and advantage of the proposed method of projecting the forcing function $f$ onto the tangent space compared to projecting the predicted state $u$ onto the constraint manifold? Projecting $u$ onto the constraint manifold seems simpler than projecting $f$ onto the tangent space, while still satisfying the constraints.
3.	The authors suggest restricting $f$ to the tangent space to satisfy the constraints. This leads to a question related to the one above: Is the original problem Eq. (1) with constraints on the state equivalent to the constrained problem Eq. (8) considered by the authors with the forcing term in the tangent space? Eq. (8) may limit the range of expressible dynamics.
4.	There is a concern that the computation of the adjoint differential and pseudoinverse in Eq. (7) would be quite difficult for general $g$.
5.	Instead of enforcing hard constraints, constraints could be incorporated into the loss function by penalizing it. For instance, this could involve adding the $L^2$ norm of $g(u)$=$(g(u)-0)$ as a regularization term to the existing NODE loss. An experimental comparison with this approach seems necessary.
6.	While hard constraints ensure that the constraints are satisfied, they are not necessarily superior to soft constraints. Hard constraints can limit the representational power of the network and may negatively impact training because of their complex computational structure. It is crucial to understand and experimentally verify the trade-off between satisfying constraints and the model's capacity.

**Questions:**

Please address the concerns mentioned in the Weaknesses above.

---

### Official Review · Reviewer_eEkq · 2024-11-03

**Soundness:** 2
**Presentation:** 2
**Contribution:** 1
**Rating:** 1
**Confidence:** 5

**Summary:**

In this paper, a method for learning differential equations while preserving conservation laws is proposed. Specifically, to preserve conservation laws, the authors project the learned vector field onto the tangent bundle of the manifold defined by the conserved quantities.

**Strengths:**

The paper is well-written and easy to read. Some experiments were conducted to support the effectiveness of the proposed method.

**Weaknesses:**

The proposed method is not novel because this method has already been proposed; the continuous-time model is shown in [1], and also the discrete-time model is shown in [2].

[1] Kasim, M.F. and Lim, Y.H. (2022) Constants of motion network, NeurIPS2022

[2] Matsubara, T. and Yaguchi, T. (2023) FINDE: Neural Differential Equations for Finding and Preserving Invariant Quantities, ICLR2023

In [1], the learned vector field is designed to be orthogonal to the gradient vectors of the conserved quantities. Precisely, the learned vector field is projected onto the tangent space at each point of the manifold defined by the conserved quantities, which is the same as the approach proposed in this paper. In [1], the QR decomposition is used for orthogonalization and hence the method of computing the projection operator is a little different from that of this paper, which uses the pseudo inverse.

In [2], exactly the same approach as this paper is proposed; in [2], the manifold defined by conserved quantities is first introduced. Then, they consider tangent bundles of this manifold, and project the leaned vector field onto the tangent space at each point. More precisely, in [2], a continuous-time model is first considered. Equation (6) in [2], which represents the continuous-time model, is completely identical to (7) in this paper. The pseudo-inverse matrix is used for projection in [2], the pseudo-inverse matrix is specifically computed though (so the model looks a little different.) In addition, in [2] a discrete-time model is also discussed. In the discrete-time model, the discrete gradient, which is a discrete version of the gradient, is considered, and the discrete tangent space is defined using the discrete gradient. The discrete-time model is essentially the projection onto this discrete tangent space.

In addition, it seems that the conserved quantities are assumed to be given in this paper; however, the methods shown in the above papers can handle cases where these quantities are unknown.

Considering the above, the contributions of this paper is quite limited.

**Questions:**

My concerns regarding this paper are as explained above.

---

### Official Review · Reviewer_1Zyj · 2024-11-03

**Soundness:** 3
**Presentation:** 4
**Contribution:** 3
**Rating:** 8
**Confidence:** 4

**Summary:**

The paper addresses the challenge of learning constrained dynamical systems in the context of neural differential equations (i.e., NDEs herafter). This term of NDEs includes the 2018 class of NODEs and generalization therefore such UDEs and bagging them together the authors are interested since indeed they allow for flexible modeling of dynamical systems by parameterizing the vector field f_\theta with neural networks. The authors observe however, that they do not inherently enforce possible known constraints that the system may have, such as conservation laws, holonomic constraints (both applying quite well in Hamiltonian systems for example), or algebraic relationships and this can lead to learned models that to different levels of severity may violate essential properties of the system, resulting in poor generalization and numerical instability.

To overcome this limitation, the authors introduce the "Projected Neural Differential Equations" (PNDEs) whose key idea is to enforce constraints by projecting the neural network vector field f_\theta onto the tangent space T_M of the constraint manifold M, which is defined by (algebraic equations) g(u) = 0. Specifically, they define the projected vector field as Proj_u (f_\theta) \in T_uM
where \mathrm{Proj}_u is the orthogonal projection operator of from T_uE onto T_uM. By integrating this projected vector field, the solutions remain on the manifold M, ensuring that the constraints are satisfied for all time. So in some sense, they try to "get rid" of the components of the vector field/neural net that would be learned but would live outside the constrained submanifold the physical system actually lives on.

The authors provide a detailed derivation of the projection operator using common decomposition techniques. They demonstrate that for an embedded submanifold M defined by smooth constraint functions g(u) the projection can be explicitly computed using the Jacobian of the latter, i.e., the constraints. This allows for efficient computation of the projected vector field during numerical integration.

To validate their approach, the authors conduct experiments on several challenging dynamical systems: the Fermi–Pasta–Ulam–Tsingou lattice system, a damped pendulum system with holonomic constraints and power grid models that incorporate power flow equations.

Compared to various existing methods they cite, such as SNDEs,  the experiments seem to verify claims that the proposed method offers exact constraint enforcement without introducing additional hyperparameters and/or suffering from stiffness issues in numerical integration which arguably distinguishes PNDEs from penalty-based methods or those that incorporate constraints as soft losses during training, which may not guarantee constraint satisfaction during inference.

Overall, the paper presents a principled and general framework for incorporating hard constraints into neural differential equations by projecting the neural network vector field onto the tangent space of the constraint manifold. This approach enhances the modeling of constrained dynamical systems, improving accuracy, generalizability, and numerical stability.

**Strengths:**

The paper presents a substantial advancement in the field of neural differential equationss given that the authors address a, indeed, crucial limitation of standard NDEs—the inability to enforce known constraints in the learned dynamical systems—which often leads to poor generalization and numerical instability. While many solutions to this have been discussed, e.g. within Hamiltonian neural nets, the strengths of the paper are several and go beyond the literature, to the best of my knowledge. c

The introduction of PNDEs is novel contribution and the authors provide a principled method to incorporate hard constraints directly into the learning process. Their approach differs from e.g., Stabilized Neural Differential Equations by ensuring that the constraints are satisfied exactly, rather than asymptotically or approximately. Using projection operators is creative and less common within "deep learning" as opposed to more traditional convex optimization.

The paper demonstrates rigorous, but bried, theoretical development and provides clear mathematical derivations. The authors derive the projection operator using the Jacobian of the defining constraint functions, and in detail explain and ensure that the projected vector field remains within the tangent space of M and they further go to propose in detail that solutions to the PNDE remain on the constraint manifold is well-founded, and the proof is succinct yet thorough. Some extra discussion as well as graphical illustration would be beneficial here! The experimental section is robust, and covers a range of systems.

In terms of clarity, I am happy to see a quite clear and well-written paper which manages to convey complex ideas effectively. That said, certain sections would be harder to follow for people with more ML/DL background but I won't count this as a limitation. Note that the motivation behind enforcing hard constraints in NDEs is very clearly articulated, and the limitations of existing methods are adequately discussed. The derivation of the projection operator is presented step-by-step (again, a graphical illustration would do miracles here), making it accessible to mathematically include readers with a background in differential geometry and dynamical systems. The experimental figures and tables are informative and enhance the understanding of the results. The experimental setup is described in sufficient detail, allowing for reproducibility (although I could not locate a link with a repo).


As mentioned earlier, incorporating hard constraints into NDEs has significant implications for modeling realistic dynamical systems that inherently possess constraints, such as conservation laws and algebraic relationships. The ability of PNDE to enforce these constraints exactly enhances the reliability and accuracy of the models, which is crucial in safety-critical applications like power grid management. However, many practicioners, especially for this example, would claim that the lack of rigorous guarantees is a problem. Returnign to the main ideas of the paper, when suitable examples are considered, improving generalization and numerical stability, PNDEs contribute to advancing the state-of-the-art in data-driven modeling of dynamical systems. This work arguably opens up new possibilities for applying NDEs to a broader class of problems where constraints play a vital role.

**Weaknesses:**

While the paper, as discussed, presents a novel and effective method for incorporating hard constraints into neural differential equations there are several areas where the work could be improved.

One of the (few) main weaknesses lies in the discussion of related work and positioning of the proposed method within the existing literature. The paper focuses primarily on comparing PNDEs to SNDEs. However, there is a rich body of research on incorporating physical constraints and conservation laws into neural network models of dynamical systems that is not adequately addressed.

For instance, the (indeed) cited HNNs [Greydanus et al., 2019] and Symplectic ODE-Net [Zhong et al., 2020] (since the author do mention inductive bias in the intro) are significant contributions that leverage the symplectic structure of Hamiltonian systems to enforce conservation of energy and other invariants. These methods learn the Hamiltonian function directly and ensure that the learned dynamics preserve the symplectic form, inherently satisfying certain physical constraints. Therefore, it's not clear to as wether the PNDEs would be relevant in systems where HNNs seem to perform very well. As a matter of fact, in recent work on learning generalized Hamiltonians using fully symplectic mappings [Choudhary et al. 2024], addresses the challenge of modeling non-separable Hamiltonian systems using implicit symplectic integrators within neural networks which should be a class of problems where previously I would had assumed PNDEs to be prime candidates to work on but its just not clear to me as to what the best approach would be in such situations. So, overall, I would prefer a more thoorough discussion here. Finally, I would be keen for the authors to portray further unerstanding of the literature of projections. For example, it is known that such projections, introduce certain symmetries. These symmetries ideally can be quotioned out in order to fascillitate easier training, see for example a similar construction in convex optimization and SDPs where the tangential projection symmetries need be addressed [Bellon et. al. 2210.08387].


While the paper provides a clear derivation of the projection operator and proves that solutions to the PNDE remain on the constraint manifold, the theoretical analysis could be strengthened. Specifically, the paper lacks a discussion on the computational complexity and scalability of the projection operation in high-dimensional systems or with complex constraints. Maybe it's too hard? From the practicioner's point of view this is important too. Given the experiments discuss power grid (we would use BnC methods normally and not gradient based methods for a number of reasons) this is important. Also, computing the projection onto the tangent space requires solving a system involving the Jacobian of the constraints, which can be computationally intensive for large-scale systems.

Moreover, the paper does not provide theoretical guarantees on the convergence or stability of the PNDEs beyond the preservation of the constraints. Are there some assumptions that can be made that would allow for an analysis of the numerical errors introduced by the projection and their impact on the overall solution accuracy? Additionally, insights into how the method performs under approximate constraints or in the presence of noise would enhance the understanding of its robustness.


Re the experimental section, while demonstrating the effectiveness of PNDEs on several systems, could be expanded to provide a more comprehensive evaluation. The experiments focus on systems where the constraint manifold is relatively straightforward to compute. It would be valuable to test PNDEs on "less trivial" systems with high-dimensional constraints or where the constraint manifold has nontrivial topology maybe?

Furthermore, the comparison is primarily with SNDEs and unconstrained NDEs. Including additional baseline methods, such as HNNs, Symplectic Neural Networks, or other constraint-enforcing techniques, would strengthen the empirical evaluation. This would provide a clearer picture of the advantages and limitations of PNDEs relative to existing approaches.

While the paper is generally well-written, certain sections could be clarified for better accessibility. The derivation of the projection operator, although mathematically rigorous, might be challenging for readers not deeply familiar with differential geometry. Providing more intuitive explanations or illustrative examples could help bridge this gap.

Additionally, the notation used in some equations, such as the use of adjoints and pseudoinverses, could be explained in more detail. Ensuring that all symbols and operations are clearly defined would improve the readability of the paper.


Another point is that the proposed method assumes that the constraints can be expressed as explicit algebraic equations and that the Jacobian of the constraints is full rank. In practice, many systems might have constraints that are implicit, differential-algebraic, or have singular Jacobians. What happens then? Discussing how PNDEs could be extended or adapted to handle such cases would enhance the significance and applicability of the work.

**Questions:**

Can the authors add some "cartoons" to ensure that readers less inclined with differential geometry can still understand the main (pictorially easy to be fair) intuition behind the paper?

Can the authors expand the discussion on relevant literature (maybe with some table comparison?) as per the "Weaknesses" section?

Can you provide insights into the computational complexity of the projection operation and discuss potential scalability issues (needs not be FOCS-style theory). Include analysis on the numerical stability and error propagation introduced by the projection maybe if possible?

Re the comment above "The experiments focus on systems where the constraint manifold is relatively straightforward to compute. It would be valuable to test PNDEs on "less trivial" systems with high-dimensional constraints or where the constraint manifold has nontrivial topology maybe? " can you maybe design such a larger instance and maybe harder topology problem? I would not decline the paper based on this but I do think it would massively strengthen the paper.

Note the unique typos I found was that no $ sign was used in a couple of T_uM instances, just make sure you fix this.

By addressing these points, the paper would be strengthened in terms of positioning within the existing literature, theoretical rigor, empirical validation, and clarity, ultimately enhancing its significance and impact on the field and would make it a super strong paper for ICLR 2025.

---

### Note · Authors · 2024-11-21

**Comment:**

We would like to thank all of the reviewers for their thorough and constructive feedback on the paper. Unfortunately, we have to agree with Reviewer eEkq that the proposed method was already derived in prior works. We were not aware of these works, and we thank the reviewer for bringing them to our attention.

While we think our experiments demonstrate the benefits of the approach in a number of new and interesting directions, the method itself is not new and we are withdrawing the paper.

Again, we would like to thank the reviewers for their time and for the thoughtful reviews.

**Withdrawal Confirmation:**

I have read and agree with the venue's withdrawal policy on behalf of myself and my co-authors.